# Anti-Inflammatory and Antinociceptive Activities of the Essential Oil of *Tagetes parryi* A. Gray (Asteraceae) and Verbenone

**DOI:** 10.3390/molecules27092612

**Published:** 2022-04-19

**Authors:** Hansel E. González-Velasco, María S. Pérez-Gutiérrez, Ángel J. Alonso-Castro, Juan R. Zapata-Morales, Perla del C. Niño-Moreno, Nimsi Campos-Xolalpa, Marco M. González-Chávez

**Affiliations:** 1Facultad de Ciencias Químicas, Universidad Autónoma de San Luis Potosí, Dr. Manuel Nava Martínez #6, Zona Universitaria, San Luis Potosí 78210, Mexico; hanseliud19@gmail.com (H.E.G.-V.); ncarmenp@uaslp.mx (P.d.C.N.-M.); 2Departamento de Sistemas Biológicos, Universidad Autónoma Metropolitana-Xochimilco, Calzada del Hueso #1100, Col. Villa Quietud, Ciudad de México 04960, Mexico; nimsicaxo@hotmail.com; 3Departamento de Farmacia, División de Ciencias Naturales y Exactas, Universidad de Guanajuato, Noria Alta S/N, Guanajuato 36050, Mexico; angeljosabad@hotmail.com (Á.J.A.-C.); mzrj@hotmail.com (J.R.Z.-M.)

**Keywords:** *Tagetes parryi*, essential oil, verbenone, anti-inflammation, antinociceptive

## Abstract

*Tagetes parryi* is a plant empirically used to treat gastrointestinal and inflammatory diseases, its essential oil (EOTP) was obtained from the aerial parts, and the composition was elucidated by GC-MS. The in vivo and in vitro anti-inflammatory activities and the antinociceptive activity of EOTP and (1*S*)-(-)-verbenone (VERB) were assessed. The major compounds identified for EOTP were verbenone (33.39%), dihydrotagetone (26.88%), and tagetone (20.8%). EOTP and VERB diminished the ear oedema induced with TPA by 93.77 % and 81.13 %, respectively. EOTP and VERB decreased inflammation in a 12-O-tetradecanoylphorbol-13-acetate (TPA) chronic model with ED_50_ = 54.95 mg/kg and 45.24 mg/kg, respectively. EOTP (15 µg/mL) inhibited the in vitro production of the pro-inflammatory mediators NO (67.02%), TNF-α (69.21%), and IL-6 (58.44%) in LPS-stimulated macrophages. In the acetic induced writhing test, EOTP and VERB showed antinociceptive effects with ED_50_ = 84.93 mg/kg and ED_50_ = 45.24 mg/kg, respectively. In phase 1 of the formalin test, EOTP and VERB showed no antinociceptive effects, whereas in phase 2, EOTP (ED_50_ = 35.45 mg/kg) and VERB (ED_50_ = 24.84 mg/kg) showed antinociceptive effects. The antinociceptive actions of ETOP and VERB were blocked with the co-administration of L-NAME. This study suggests that EOTP and VERB might be used in the treatment of pain and inflammatory problems.

## 1. Introduction

Inflammation is a biological response to different stimuli, such as pathogens, physical trauma, and chemical irritants [1]. Inflammation can be acute, as in a short-term process, or chronic as in long-term one [2]. Acute inflammation is characterized by the activation of innate immune cells such as neutrophils and macrophages that move from the blood into the injured site. Chronic inflammation mainly involves the activation of the monocyte into macrophages, and it is characterized by the simultaneous destruction and healing of the injured site [3,4].

Inflammation is mediated by different biochemical cascades, such as the activation of phospholipase A_2_ in the cell membrane, followed by the release of arachidonic acid derivatives and many other inflammatory mediators, such as cytokines, leukotrienes, serotonin, and histamine [2]. Many anti-inflammatory drugs have adverse effects [5,6]. Therefore, it is necessary to search for new active compounds with anti-inflammatory activity with fewer adverse effects, and plants are a potential source of new anti-inflammatory molecules [7]. 

In Mexican traditional medicine, there are aromatic plants with several pharmacological effects, including antimicrobial, cytotoxic, spasmolytic, antinociceptive, and anti-inflammatory [8]. The *Tagetes* genus (Asteraceae) is composed of at least 55 plant species, which are aromatic medicinal plants [9]. The pharmacological activity of different plant extracts, compounds, and essential oils obtained from species of the *Tagetes* genus has been studied, and several of these essential oils (EOs) have antimicrobial, antioxidant, cytotoxic, antinociceptive, and anti-inflammatory activities [10,11,12]. The EOs of *Tagetes minuta* inhibited the expression of mRNA for iNOS and different pro-inflammatory cytokines, such as TNF-α [13], whereas the EO of *T. lucida* diminished the production of NO and PGE_2_ [12]. The EO of *Tagetes lucida* showed antinociceptive actions in rats [14].

*Tagetes parryi* is a medicinal plant endemic to San Luis Potosí, commonly known as cincoyaga, which is used for the empirical treatment of gastrointestinal disorders and inflammatory problems. This study aimed to obtain the essential oil of *Tagetes parryi*, determined its chemical composition, and evaluated the anti-inflammatory and antinociceptive effects, using in vivo and in vitro models, of the essential oil and its main constituent verbenone.

## 2. Results and Discussion

### 2.1. Chemical Composition of EOTP

EOTP was analysed, and its chemical composition was determined by gas chromatography-mass spectrometry (GC-MS). Twenty-one compounds were identified, representing 92.02% of the total (Table 1, Appendix A).

Most of the 21 compounds were monoterpenes and sesquiterpenes. The main compounds were verbenone (33.39%), dihydrotagetone (26.88%), and tagetone (20.8%). These results were different from those reported by Días-Cedillo et al. [15], who found seven compounds, α-terpineol and eugenol to be present in both essential oils from *Tagetes parryi*, and the differences in the composition of the two essential oils may be due to the season and place of the collection. However, the EOTP has many common compounds previously reported in other species of *Tagetes* with the difference in proportions of dihydrotagetone and tagetone [16,17].

### 2.2. Acute Toxicity

In this assay, all mice survived even at the highest dose of EOTP (2000 mg/kg p.o.). Therefore, the LD_50_ of EOTP is higher than 2000 mg/kg. In the EOTP acute toxicity evaluation, the mice did not show behavioural changes, toxicity signs, or death after the administration of EOTP at a dose of 2000 mg/kg. 

### 2.3. Anti-Inflammatory Activity 

The in vivo anti-inflammatory activity of EOTP and VERB was determined in the TPA-induced mouse ear oedema at a dose of 2 mg/ear. EOTP and VERB diminished oedema by 93.77%, and 81.13%, respectively. The anti-inflammatory effect of EOTP and VERB was comparable to that of indomethacin (81.3%). EOTP and VERB diminished ear oedema induced by multiple applications of TPA, and the ED_50_ values obtained for VERB and EOTP were 54.95 and 45.24 mg/kg, respectively. The oedema inhibition at a dose of 100 mg/kg EOTP was similar to the anti-inflammatory activity obtained with 5 mg/kg indomethacin (IND) (Table 2). 

Inflammation produced in the ear oedema induced with TPA applied topically is mediated by protein kinase C, the stimulation of phospholipase A_2,_ and cyclooxygenase. The results suggest that the anti-inflammatory activity of EOTP and VERB is due to the inhibition of these inflammatory mediators [18].

EOTP and VERB were tested at different doses on chronic TPA-induced oedema. TPA applied to mouse ears in multiple applications produces infiltration of inflammatory cells, such as leukocyte and epidermal hyperplasia; therefore, there is an increase in ear oedema size [19]. EOTP and VERB diminished ear oedema induced by multiple applications of TPA in a dose-dependent manner, this result suggests that the EOTP and VERB could diminish the cellular infiltration. 

### 2.4. Cell Viability 

The effect of EOTP (1 to 200 µg/mL) on the cell viability of J447A.1 was determined with a 3-(4,5-dimethylthiazol-2-yl)-2,5-diphenyl tetrazolium bromide (MTT) assay. The IC_50_ obtained was 30.78 µg/mL. Therefore, a concentration of 15 µg/mL EOTP was used in subsequent experiments. The EO of *Tagetes minuta* showed an IC_50_= 95 µg/mL in the cell viability of J774A.1 macrophages [13].

### 2.5. NO and Cytokine Production

When the macrophages were treated with 5 µg/mL LPS for 24 h, the NO concentration was increased (Figure 1). EOTP at a concentration of 15 µg/mL inhibited the NO release by 67.02%, with a similar effect to that obtained with IND at a concentration of 17.1 µg/mL (58.12% inhibition). NO has an important role in inflammation pathogenesis, and its release induces the activation of cytokines, chemokines that induce vasodilatation, and the formation of reactive species of nitrogen, producing tissue damage [20]. The compounds that inhibited NO production could be useful in the treatment of inflammatory problems [21].

EOTP inhibited the production of the pro-inflammatory cytokines TNF-α (69.21%) and IL-6 (58.44%) in LPS-stimulated macrophages. TNF-α is involved in the acute inflammatory process, it plays an important role in the regulation of other pro-inflammatory mediators, such as IL-6, and it also keeps inflammation localized [22]. However, the increase in levels of TNF-α could be harmful [23]. The interleukin IL-6 is responsible for the induction and perpetuation of the inflammatory process and amplifies the inflammatory cascade, producing tissue damage [24]. Thus, it is important to find agents that inhibit pro-inflammatory cytokines because they might effective in controlling different inflammatory diseases. The EOs of *Tagetes minuta* (1–50 ug/mL) and *Tagetes lucida* (200 ug/mL) decreased the mRNA expression of TNF-α and NO in LPS-stimulated macrophages [12,13]. These findings indicate that EOs from the *Tagetes* genus contain compounds with anti-inflammatory actions.

### 2.6. Antinociceptive Activity

In the acetic acid-induced writhing model, EOTP and VERB reduced the number of abdominal contortions in a dose-dependent manner (*p* < 0.05). The ED_50_ values for EOTP and VERB were 84.93 ±5.64 and 63.80 ± 3.60, respectively. EOTP and VERB at a dose of 200 mg/kg showed an inhibition in the nociception (64.62% and 82.76%, respectively) similar to that obtained with 100 mg/kg naproxen sodium (NPX) (71.03%) (Table 3).

The ED_50_ values of EOTP and VERB in phase 1 of the formalin test were 181.9 ± 16 mg/kg and 97.0 ± 4.85 mg/kg, respectively, and the ED_50_ values for phase 2 were 35.45 ± 7.5 mg/kg and 24.84 ± 8.6 mg/kg, respectively (Figure 2). These results showed that EOTP and VERB have significant antinociceptive effects (*p* < 0.05) and show dose-dependent behaviour in phase 2, and at a dose of 200 mg/kg, EOTP showed comparable results with the control tramadol (TRD) (10 mg/kg i.p.). The EO of *Tagetes lucida* showed antinociceptive activity in a dose-dependent manner in the writhing test and the formalin test in rats. The EO of *Tagetes lucida* showed an ED_50_ = 14.54 mg/kg (i.p.) in the first phase of the formalin test [14]. The EO of *Dracocephalum kotschyi* has verbenone as its principal component and showed antinociceptive effects with ED_50_ = 61.61 mg/Kg in the mouse writhing test [25].

#### Determination of the Antinociceptive Mechanism

The mechanism of the antinociceptive activity of EOTP (100 mg/kg) and VERB (24.84 mg/kg) was determined with the formalin test using antagonists. EOTP and VERB were administered 15 min after the administration of antagonists naloxone (NAL), glibenclamide (GLI), and Nω-nitro-L-arginine methyl ester (L-NAME). In phase 1, no loss of the antinociceptive effect was observed (Figure 3). In phase 2, the antinociceptive activities of EOTP and VERB were reverted in the presence of L-NAME; therefore, the mechanism of action could be through the inhibition of the NO pathway.

Antinociception was evaluated in two different models: the acetic acid-induced writhing test and the formalin test. The injection of acetic acid causes increases in the levels of prostaglandins, arachidonic acid, serotonin, bradykinin, and histamine which trigger the writhing response [26,27]. EOTP and VERB showed a dose-dependent effect on the behaviour in this test. The results obtained at the highest dose of ETOP (200 mg/kg p.o.) were comparable with the positive control (NPX, 100 mg/kg p.o.).

The formalin test has a biphasic effect: in phase 1, there is irritation in the hind paw due to activation of C-fiber nociceptors (neurogenic pain). In phase 2 (inflammatory pain), many pro-inflammatory mediators such as cytokines, prostaglandins, and bradykinin are released [28,29]. In phase 2, EOTP and VERB decreased licking time in a dose-dependent manner. ETOP (200 mg/kg) exerted similar antinociceptive effects compared to those shown in the positive control (TRD, 10 mg/kg i.p.). These results suggest that EOTP and VERB inhibited the inflammatory process derived from formalin injection. To elucidate their antinociceptive mechanism, EOTP (100 mg/kg p.o.) and VERB (24.84 mg/kg p.o.) were co-administered with NAL (2 mg/kg i.p.), GLI (10 mg/kg i.p.) or L-NAME (20 mg/kg i.p.). The antinociceptive effects of EOTP and VERB were reverted in the presence of L-NAME, suggesting that the possible mechanism of antinociceptive action of EOTP and VERB is the inhibition of the NO pathway. NO is involved in the activation of glutamate channels and the release of substance P, two neurotransmitters that induce pain, and the overproduction of NO promotes the release of pro-inflammatory mediators, leading to inflammatory and nociceptive reactions in peripheral nerve injury [30,31,32]. These results confirm the involvement of nitric oxide in the antinociception activity shown by the terpene VERB.

## 3. Materials and Method

### 3.1. Chemicals

Nω-nitro-L-arginine methyl ester (L-NAME), glibenclamide (GLI), naloxone (NAL), tramadol (TRD), naproxen sodium (NPX), (1*S*)-(-)-verbenone (VERB), foetal bovine serum (FBS), Dulbecco’s modified eagle medium (DMEM), antibiotics, 3-(4,5-dimethylthiazol-2-yl)-2,5-diphenyl tetrazolium bromide (MTT), indomethacin (IND), diclofenac sodium (DICLO), polyvinylpyrrolidone (PVP), Griess reagent (GR), and 12-O-tetradecanoylphorbol-13-acetate (TPA) were purchased from Millipore Sigma (St Louis, MO, USA).

### 3.2. Plant Material

*Tagetes parryi* was collected in November 2013 in the community of Agua Blanca, in the municipality of Villa de Zaragoza in San Luis Potosí State, Mexico. José Garcia Pérez identified the plant material. A voucher specimen was deposited in the Herbarium Isidro Palacios of the Universidad Autónoma of San Luis Potosí (SPLM31975).

### 3.3. Obtention of the Essential Oil

The essential oil *T. parryi* was obtained from 1650 g of the aerial parts of the fresh plant by hydrodistillation in a simple distillation apparatus. The essential oil was extracted with dimethyl ether and concentrated under reduced pressure at 20 °C to eliminate the solvent and then treated with anhydrous sodium sulphate to eliminate any remaining water, resulting in 8.9 g of essential oil with a yield of 0.54%. Afterward, the essential oil was stored at 5 °C.

### 3.4. Analysis of EOTP

The chemical composition of EOTP was determined by GC-MS using a chromatograph model 6890N Network GC System connected to a selective mass detector model 5973 Network (MSD) (Agilent Technologies, Wilmington, DE, USA). The separation was performed with an HP-5MS capillary column (30 m in length, 0.25 mm internal diameter, and 0.25 µm film width) (J&W, Folsom, CA, USA). The injector was operated in the splitless mode at 240 °C and helium was used as the carrier gas at 1 mL/min. The chromatographic method began at a temperature of 50 °C (3 min) with an increment rate of 3 °C/min to 240 °C (2 min).

The MSD was operated at 70 eV in the mass range of 15 to 800 atomic mass units, the ion source was set at 150 °C, and the transfer line was at 250 °C. The compounds were identified by comparing their mass spectrum with those reported in the Willey 09 and NIST11 libraries. In addition, the Kovak index was calculated for each peak with reference to the n-alkane standards (C6-C26) running under the same conditions.

### 3.5. Animals

CD1 male mice weighing 25–30 g were obtained from the animal facility of the Universidad Autónoma Metropolitana Unidad Xochimilco for the inflammatory evaluation, and BALB/c mice were obtained from the División de Ciencias Naturales y Exactas, Universidad de Guanajuato for the antinociceptive test. The animals were housed in isolated cages at 24 °C under a light-dark cycle of 12:12 h.

### 3.6. Acute Toxicity of EOTP

The acute toxicity evaluation of EOTP was performed according to the Lorke methodology [33]. Mice were administered with EOTP (10–2000 mg/kg p.o.). After treatment, behavioural changes, toxicity signs, and mortality were recorded in the mice daily for 72 h. LD_50_ values were calculated by linear regression. The LD_50_ of VERB was previously determined in rats, LD_50_ = 3400 mg/kg for males and LD_50_ = 1800 mg/kg for females [34]. For this reason, the acute toxicity of VERB was not evaluated.

### 3.7. Anti-Inflammatory Activity

#### 3.7.1. 12-O-Tetradecanoylphorbol-13-acetate (TPA)-Induced Ear Oedema

De Young et al. [35] previously described the procedure of TPA-induced mouse ear oedema. Mice were separated into four different groups (*n* = 8 per group). The right ear received a topical application of a solution containing 2.5 µL of TPA in 25 µL of acetone. Thirty minutes after the TPA application, the following treatments were administered in the right ear, at a dose of 2 mg/ear: group 1 indomethacin (IND), group 2 EOTP, group 3 VERB, group 4 acetone (vehicle) was administered in the left ear.

After 6 h, animals were sacrificed by cervical dislocation. A central section of 6 mm from both ears was obtained and weighed to determine the percent inhibition as follows:(1)$$\%\;\mathrm{Inhibition}=(\frac{\left(W_{t}-W_{nt}\right)\mathrm{control}-\left(W_{t}-W_{nt}\right)\mathrm{treated}}{\left(W_{t}-W_{nt}\right)\mathrm{control}})100$$*W_t_*: weight of treated ear, *W_nt_*: weight of a non-treated ear.

#### 3.7.2. Chronic Anti-Inflammatory Activity of TPA-Induced Mouse Ear Oedema

The model of chronic TPA-induced ear oedema in mice was carried out as described previously [36]. Each group consisted of eight mice. A solution of 2.5 μL of TPA/ear in 25 μL of acetone was topically administered. After thirty minutes, the mice were orally administered the following samples: vehicle (saline solution), EOTP (25, 50, 100, or 200 mg/kg p.o.), or VERB (12.5, 25, 50, or 100 mg/kg p.o.). The doses were chosen according to the reference drug, and IND (5 mg/kg) was used as a positive control. Mice were orally administered the treatment and TPA every 48 h for 10 days. At the end of the experiment, the mice were sacrificed by cervical dislocation, and 6 mm discs were obtained from both ears. The formula described above was used to obtain the percent inhibition.

### 3.8. Antinociceptive Activity

Acetic acid-induced writhing test

This assay was performed according to Koster [37]. Mice were divided into four different groups (*n* = 8 per group). One hour before the injection with acetic acid, mice were administered a dose of 100 mg/kg (p.o.) NPX; EOTP or VERB, at different doses (10, 50, 100, or 200 mg/kg), and the control group received saline solution. One hour after treatment administration, animals were injected with 10 mL/kg (i.p.) of an acetic acid solution (1% *v/v*). Next, the mice were individually placed into a glass cylinder and writhing was counted for 30 min. The ED_50_ was obtained by regression analysis.

Formalin test

The formalin test was performed according to Hunskaar and Hole [28]. One hour before the administration of formalin, mice (*n* = 8 per group) received one of the following treatments: vehicle (saline solution), EOTP, or VERB at different doses (10–200 mg/kg p.o.), or TRD (10 mg/kg i.p.). Mice were injected with 30 µL of 3% formalin in isotonic saline solution into the sub-plantar space of the right hind paw and then were individually placed in plastic cylinders. The duration of paw licking was recorded from 0–15 min (first phase) and 15–45 min (second phase). The ED_50_ was obtained by linear regression analysis.

Evaluation of the possible mechanisms of action in the antinociceptive activity

Balb/c mice (*n* = 8 per group) were administered with EOTP (100 mg/kg p.o.) or VERB (24.84 mg/kg p.o.), and after 15 min, the animals were administered with NAL (2 mg/kg i.p.), L-NAME (20 mg/kg i.p.), or GLI (10 mg/kg i.p.). Formalin was injected 30 min after the administration of the compounds, and the assay was carried out as described above.

### 3.9. Cell Viability

The MTT assay was used to evaluate the cytotoxicity of EOTP [38]. Murine J447A.1 macrophage were seeded (1 × 10^3^ cells/well) in a 96-well plate and treated with different concentrations of EOTP (1, 5, 10, 25, 50, 100, or 200 µg/mL) or vehicle (0.9% PBS) for 24 h. Non-treated cells (0.9% PBS) were considered as the control, representing 100% viability. After this time, 10 µL of a solution of MTT (5 mg/mL) was added to each well and incubated for 4 h at 37 °C and 5% CO_2_. After removing the medium, the formazan crystals were dissolved in 100 µL of DMSO. The optical density (OD) was measured at 540 nm on a BioRad ELISA microplate reader. Six replicate wells were used to determine viability using the formula described as follows:% viability = (Abs. treated Cells/Abs. control cells) × 100(2)

The IC_50_ was calculated by regression analysis.

### 3.10. NO and Cytokine Evaluation

J447A.1 cells were treated with LPS (5 µg/mL), EOTP (15 µg/mL), or IND (17.1 µg/mL), and non-treated cells were employed as a negative control vehicle (0.9% PBS). Cells were seeded and incubated in 96 well plates at 5 × 10^5^ cells/well under standard conditions for 24 h. The supernatant was collected for the quantification of NO [39], TNF-α, and IL-6. For the determination of NO production, 100 µL of supernatant were mixed with 100 µL of Griess reagent (1% sulphanilamide in 5% H_2_PO_4_ and 0.1% naphthylethylene diamide dihydrochloride solution). After 30 min of incubation at 37 °C, the absorbance was measured at 540 nm to obtain the OD. A 100% nitric oxide production was considered for the LPS group.

The levels of IL-6 and TNF-α were measured using a commercial ELISA kit according to the manufacturer’s instructions (Peprotech, London, UK). The OD was measured in a microplate reader at 405 nm with wavelength correction set to 650 nm.

### 3.11. Statistical Analysis

All data are expressed as the mean ± S.E.M., and the analysis was performed using a Student’s *t*-test (*p* < 0.05), and ANOVA with post hoc Dunnett’s test (*p* < 0.05) was considered an indication of significance. In this research, GraphPad Prism 6 was used.

## 4. Conclusions

EOTP has anti-inflammatory activity on in vivo and in vitro models, and this activity may be attributed to VERB and other compounds in the essential oil. The EOTP anti-inflammatory effect is due to inhibition of the production of pro-inflammatory cytokines. We observed antinociceptive activity due to the possible inhibition of the NO pathway. The acute toxicity was very low for EOTP. These results suggest that EOTP and VERB might be candidates to develop new phytomedicine to treat pain and inflammatory problems.

## Figures and Tables

**Figure 1 molecules-27-02612-f001:**
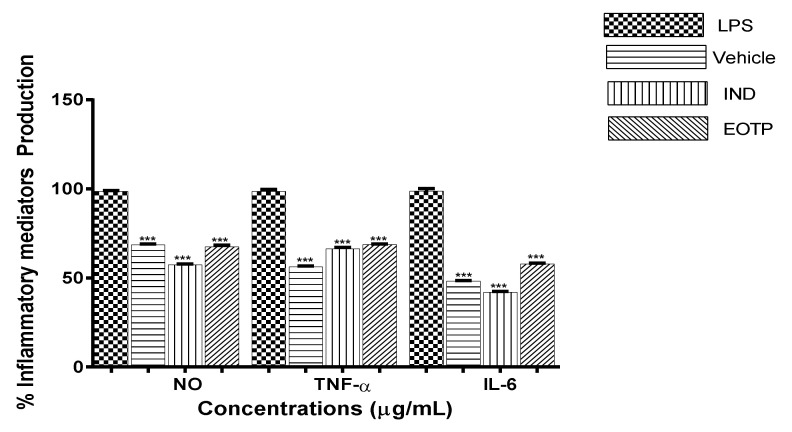
Effect of EOTP on the levels of NO, TNF-α, and IL-6 in macrophages stimulated with LPS. The results are the mean of three determinations ± SEM. *** *p* < 0.001 vs. vehicle.

**Figure 2 molecules-27-02612-f002:**
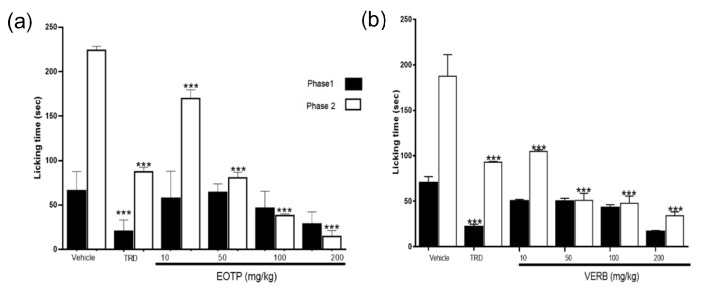
(**a**) The antinociceptive effects of EOTP in the formalin test at doses of 10, 50, 100, 200 mg/kg p.o. (**b**) The antinociceptive effects of VERB in the formalin test at different doses (10, 50, 100, 200 mg/kg p.o.) are shown in the formalin test’s two phases. TRD (10 mg/kg i.p.) was the negative control, whereas the vehicle was saline solution p.o. The figure shows the two phases of the formalin test. The experimental results were compared against the vehicle’s values expressed as the mean ± SEM. A significant difference (***) was determined at *p* < 0.05.

**Figure 3 molecules-27-02612-f003:**
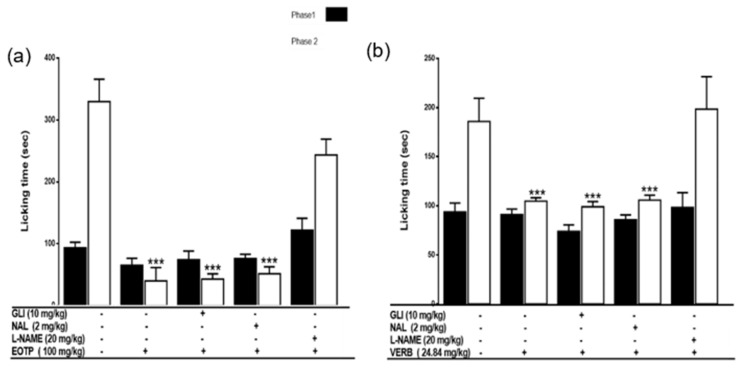
The mechanism of action of the antinociceptive effects of EOTP and VERB was evaluated by a formalin pain biphasic model in the three graphics that show both phase 1 and phase 2. (**a**) The mechanism of action of EOTP (100 mg/kg). (**b**) Determination of the antinociception mechanism of action of VERB (24.8 mg/kg). Additional groups of mice received an inhibitor such as L-NAME (20 mg/kg, an inhibitor of nitric oxide synthase and guanylate cyclase), GLI (20 mg/kg, a K^+^ channel sensitive to ATP) or NAL (2 mg/kg, an antagonist of opioid receptors). The data represent two different experiments (*n* = 8). The results represent the mean ± SEM. (***) represents *p* < 0.05 compared with the control group.

**Table 1 molecules-27-02612-t001:** The chemical composition of EOTP.

Compound	Rt (Min)	%	RI_R_	RI_E_
3-Hexenol-1-ol	5.66	0.16	838	806.3
β-Phellandrene	10.03	0.34	964	957.0
β-Pinene	10.13	0.33	961.7	960.3
β-Myrcene	10.86	0.20	979	985.6
α-Phellandrene	11.40	0.23	997	1000.0
Eucalyptol	12.63	1.45	1023	1028.2
*trans*-β-Ocimene	13.06	2.01	1034	1037.1
Dihydrotagetone	13.87	26.88	1055	1054.1
Chrysanthenone	17.20	0.38	1099	1123
Neo-allo-ocimene	17.46	0.23	1131	1128.4
Tagetone	18.70	20.8	1124	1153
Terpinene-4-ol	19.75	0.13	1161	1188.5
α-Terpineol	20.42	0.56	1172	1188.5
2-Ethylidene-6-methyl-3,5-heptadienal	21.22	0.40	1182	1205
Verbenone	22.95	33.39	1228	1242.4
Thymol	23.47	0.18	1266	1253.6
Isopiperitenone	24.32	2.55	1249	1271.9
Eugenol	29.96	1.42	1392	1393.3
Caryophyllene	31.06	0.31	1424	1418
*p*-Cresol	33.12	0.16	1503.9	1474.1
Elemol	36.61	0.11	1535	1551.2
Total		92.02		

Retention time (Rt), retention indexes in the literature (RI_R_), and retention indexes calculated (RI_E_).

**Table 2 molecules-27-02612-t002:** Anti-inflammatory activity of EOTP, and VERB, on ear oedema-induced by multiple TPA applications in mice.

Compound/Dose	% Inhibition of Inflammation
Control	-
IND 8 mg/kg	64.67 ± 2.16 ***
EOTP 25 mg/kg	30.37 ± 2.41
EOTP 50 mg/kg	45.92 ± 3.26
EOTP 100 mg/kg	68.54 ± 1.92 ***
EOTP 200 mg/kg	80.40 ± 3.93 ***
VERB 12.5 mg/kg	36.95 ± 2.96
VERB 25 mg/kg	42.32 ± 3.84
VERB 50 mg/kg	46.16 ± 2.57
VERB 100 mg/kg	62.91 ± 3.95 ***

The data are expressed as the means ± SEM. (*n* = 8). (***) represents *p* < 0.05 vs. IND.

**Table 3 molecules-27-02612-t003:** The antinociceptive activity of EOTP and VERB in the acetic acid-induced writhing model.

Compound/Dose	% Inhibition of Antinociception
Control	-
NPX 100 mg/kg	71.03 ± 3.30 ***
EOTP 10 mg/kg	27.48 ± 7.20
EOTP 50 mg/kg	39.86 ± 4.61
EOTP 100 mg/kg	48.70 ± 8.37
EOTP 200 mg/kg	64.62 ± 2.37 ***
VERB 10 mg/kg	12.76 ± 4.88
VERB 50 mg/kg	36.55 ± 3.61
VERB 100 mg/kg	56.55 ± 3.07
VERB 200 mg/kg	82.76 ± 2.18 ***

The data are expressed as the means ± SEM. (***) represents *p* < 0.05, NPX vs. EOTP or VERB (*n* = 8).

## Data Availability

The chromatograms obtained in this study are available in the Appendix A.

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
