# Peer review of "Anti-Inflammatory and Antinociceptive Activities of the Essential Oil of Tagetes parryi A. Gray (Asteraceae) and Verbenone"

_molecules, 2022, doi:10.3390/molecules27092612_

Round 1

Reviewer 1 Report

The manuscript "Anti-inflammatory and antinociceptive activities of the essential oil of Tagetes parryi A. Gray (Asteraceae) and verbenone" is is very interesting. The introduction is well written and clearly describes the subject to be covered.
The results and discussion are well presented and compared with the literature.

Reviewer 2 Report

The manuscript "Anti-inflammatory and antinociceptive activities of the essential oil of Tagetes parryi A. Gray (Asteraceae) and verbenone" has a methodical research plan and an interesting topic.

The research is well organized, the purpose is clear and the tests are relevant. However, my recommendation for the authors is to improve the conclusion section.  Are other species with the same activities? As the authors mentioned in the introduction, they are, so, if possible, I recommend making a comparison between them. Even though the authors have made some observation after presenting the result for each test, an overall discussion should be more interesting and would make the article more useful for the readers.

In terms of writing accuracy please keep in mind that the results are presented before the methods, so even though you have mentioned each compound with the appropriate abbreviation in the methods section it appears first in earlier. Make sure that each abbreviation is explained at the first appearance in the text.

Reviewer 3 Report

The essential oil of the plant Tagetes parryi was investigated previously, but it was with different chemical composition.

the authors should cite the previous works on essential oil and they are 

Díaz-Cedillo, F. and Serrato-Cruz, M.A., 2011. Composición del aceite esencial de Tagetes parryi A. Gray. Revista fitotecnia mexicana34(2), pp.145-148.

Please indicate the factors that could contribute to the different compositions "geographical position, altitude, etc.." 

the authors have studied the essential oil and its major constituent verbenone and this is the novelty of this work. the tests performed are logical and consistent. however, some other plants have verbenone as their main constituent, do these plants have the pharmacological properties as Tagetes parryi?

Why the authors did not investigate the same pharmacological effect of dihydrotagetone and tagetone which have high percentages in the essential oil 26.88% and 20.8 % respectively.
